# 3D-GOI: 3D GAN Omni-Inversion for Multi-faceted and Multi-object Editing

## Abstract

The current GAN inversion methods typically can only edit the appearance and shape of a single object and background while overlooking spatial information. In this work, we propose a 3D editing framework, 3D-GOI to enable multifaceted editing of affine information (scale, translation, and rotation) on multiple objects. 3D-GOI realizes the complex editing function by inverting the abundance of attribute codes (object shape/appearance/scale/rotation/translation, background shape/appearance, and camera pose) controlled by GIRAFFE, a renowned 3D GAN. Accurately inverting all the codes is challenging, 3D-GOI solves this challenge following three main steps. First, we segment the objects and the background in a multi-object image. Second, we use a custom Neural Inversion Encoder to obtain coarse codes of each object. Finally, we use a round-robin optimization algorithm to get precise codes to reconstruct the image. To the best of our knowledge, 3D-GOI is the first framework to enable multifaceted editing on multiple objects. Both qualitative and quantitative experiments demonstrate that 3D-GOI holds immense potential for flexible, multifaceted editing in complex multi-object scenes.

## 1 Introduction

With the development of generative 3D models, researchers are becoming increasingly interested in generating and editing 3D objects to enhance the automation of multi-object scene generation. However, most existing works are limited to generating and editing a single object, such as 3D face generation (Chan et al., 2022) and synthesis of facial viewpoints (Yin et al., 2022). There are few methods for generating multi-object 3D scenes while editing such scenes remains unexplored. In this paper, we propose 3D-GOI to edit images containing multiple objects with complex spatial geometric relationships. 3D-GOI not only can change the appearance and shape of each object and the background, but also can edit the spatial position of each object and the camera pose of the image as shown by Figure 1.

Existing 3D multi-object scenes generation methods can be mainly classified into two categories: those based on Generative Adversarial Networks (GANs) (Goodfellow et al., 2020) and those based on diffusion models (Ho et al., 2020), besides a few based on VAE or Transformer (Yang et al., 2021; Arad Hudson & Zitnick, 2021). GAN-based methods, primarily represented by GIRAFFE (Niemeyer & Geiger, 2021) and its derivatives, depict complex scene images as results of multiple foreground objects, controlled by shape and appearance, being subjected to affine transformations (scaling, translation, and rotation), and rendered together with a background, which is also controlled by shape and appearance, from a specific camera viewpoint. On the other hand, diffusion-based methods (Lin et al., 2023) perceive scene images as results of multiple latent NeRF (Metzer et al., 2022), which can be represented as 3D models, undergoing affine transformations, optimized with SDS (Poole et al., 2022), and then rendered from a specific camera viewpoint. Both categories inherently represent scenes as combinations of multiple codes. To realize editing based on these generative methods, it's imperative to invert the complex multi-object scene images to retrieve their representative codes. After modifying these codes, regeneration can achieve diversified editing of complex images. However, most of the current inversion methods study the inversion of a single code based on its generation method, yet the inversion of multiple codes in complex multi-object scenes is largely overlooked. Each multi-object image is the entangled result of multiple codes, to invert all codes from an image requires precise disentangling of the codes which is extremely diffi-

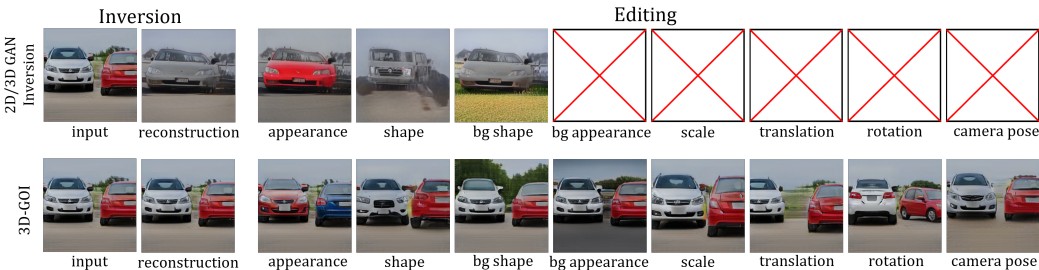

Figure 1: The first row shows the editing results of traditional 2D/3D GAN inversion methods on multi-object images. The second row showcases our proposed 3D-GOI, which can perform multifaceted editing on complex images with multiple objects. 'bg' stands for background. The red crosses in the upper right figures indicate features that cannot be edited with current 2D/3D GAN inversion methods.

cult. Moreover, the prevailing inversion algorithms (for single code) primarily employ optimization approaches. Attempting to optimize all codes simultaneously often leads to chaotic optimization directions, preventing accurate inversion outcomes.

In the face of these challenges, we propose 3D-GOI a framework capable of addressing the inversion of multiple codes, aiming to achieve a comprehensive inversion of multi-object images. Given the current open-source code availability for 3D multi-object scene generation methods, we have chosen GIRAFFE (Niemeyer & Geiger, 2021) as our generative model. In theory, our framework can be applied to other generative approaches as well.

We address this challenge as follows. First, we categorize different codes based on object attributes, background attributes, and pose attributes. Through qualitative verification, we found that segmentation methods can roughly separate the codes pertaining to different objects. For example, the codes controlling an object's shape, appearance, scale, translation, and rotation predominantly relate to the object itself. So during the inversion process, we only use the segmented image of this object, which can reduce the impact of the background and other objects on its attribute codes.

Second, we get the codes corresponding to attributes from the segmented image. Inspired by the Neural Rendering Block in GIRAFFE, we design a custom Neural Inversion Encoder network to coarsely disentangle and estimate the values of various attribute codes.

Finally, we obtain precise values for each code through optimization. We found that optimizing all codes simultaneously tends to get stuck in local minima. Therefore, we propose a round-robin optimization algorithm that employs a ranking function to determine the optimization order for different codes. The algorithm enables a stable and efficient optimization process for accurate image reconstruction. Our contributions can be summarized as follows.

- To our knowledge, we are the first to propose a multi-code inversion framework in generative models, achieving multifaceted editing of multi-object images.
- We introduce a three-stage inversion process: 1) separate the attribute codes of different objects via the segmentation method; 2) obtain coarse codes of the image using a custom Neural Inversion Encoder; 3) optimize the reconstruction using a round-robin optimization strategy.
- Our method outperforms state-of-the-art methods on multiple datasets on both 3D and 2D tasks.

## 2 PRELIMINARY

GIRAFFE (Niemeyer & Geiger, 2021) represents individual objects as a combination of feature field and volume density. Through scene compositions, the feature fields of multiple objects and the background are combined. Finally, the combined feature field is rendered into an image using volume rendering and neural rendering. The details are described as follows.

For a coordinate $x$ and a viewing direction $d$ in scene space, the affine transformation $T(s, t, r)$ (s represents scale, t represents translation, r represents rotation) is used to transform them back into the

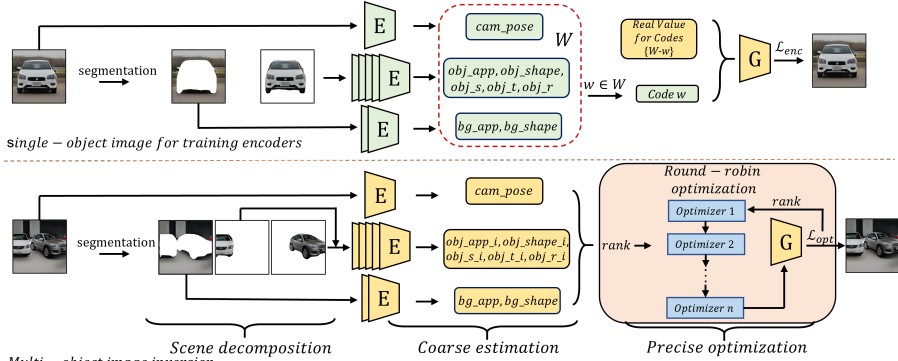

Figure 2: The overall framework of 3D-GOI. As shown in the upper half, the encoders are trained on single-object scenes, each time using $L_{enc}$ to predict one $w, w \in W$, while other codes use real values. The lower half depicts the inversion process for the multi-object scene. We first decompose objects and background from the scene, then use the trained encoder to extract coarse codes, and finally use the round-robin optimization algorithm to obtain precise codes. The green blocks indicate required training and the yellow blocks indicate fixed parameters.

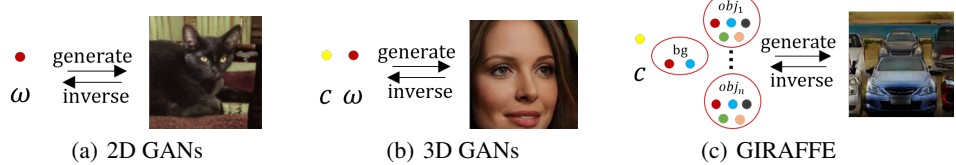

|  (a) 2D GANs | (b) 3D GANs | (c) GIRAFFE |

Figure 3: Figure (a) represents the typical 2D GANs and 2D GAN Inversion methods, where one latent encoding corresponds to one image. Figure (b) represents the typical 3D GANs and 3D GAN Inversion methods, which usually have an additional camera pose code c. Both of these methods can only generate and invert single objects. Figure (c) represents GIRAFFE, which can generate complex multi-object scenes. Each object is controlled by appearance, shape, scale, translation, and rotation, while the background is controlled by appearance and shape. Similarly, c controls the camera pose, so there are generally (5n+3) codes, far more than the number of codes in a typical GAN. Therefore, inverting it is a very challenging task.'bg' means background and 'obj' means object.

object space of each individual object. Following the implicit shape representations used in Neural Radiance Fields (NeRF) (Mildenhall et al., 2021), a multi-layer perceptron (MLP) $h_\theta$ is used to map the transformed $x$ and $d$, along with the shape-controlling code $z_s$ and appearance-controlling code $z_a$, to the feature field $f$ and volume density $\sigma$ as expressed below:

$$(T(s, t, r; \boldsymbol{x})), T(s, t, r; \boldsymbol{d})), \boldsymbol{z_s}, \boldsymbol{z_a}) \xrightarrow{h_\theta} (\sigma, \boldsymbol{f}). \tag{1}$$

Then, GIRAFFE defines a Scene Composite Operator: at a given coordinate $x$ and viewing direction $d$, the overall density is the sum of the individual densities (including the background). The overall feature field is represented as the density-weighted average of the feature field of each object, as expressed below:

$$C(\boldsymbol{x}, \boldsymbol{d}) = (\sigma, \frac{1}{\sigma} \sum_{i=1}^{N} \sigma_i \boldsymbol{f_i}), where \quad \sigma = \sum_{i=1}^{N} \sigma_i, \tag{2}$$

where N denotes the background plus (N-1) objects.

The rendering phase is divided into two stages. Similar to volume rendering in NeRF (Mildenhall et al., 2021), given a pixel point, the rendering formula is used to calculate the feature field of this pixel point from the feature fields and the volume density of all sample points in the direction of a camera ray direction. After calculating for all pixel points, a feature map is obtained. Neural rendering (Upsampling) is then applied to get the rendered image. Please refer to the Appendix B for the detailed preliminary and formulas.

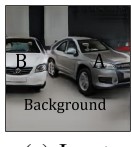 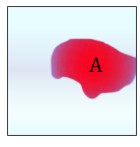 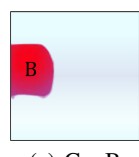 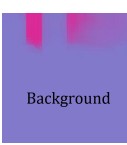

(a) Input        (b) Car A        (c) Car B        (d) Background

Figure 4: Scene decomposition. (a) is the input image. (b) is the feature weight map of car A, where the redder regions indicate a higher opacity for car A and the bluer regions indicate lower opacity. Similarly, (c) is the feature weight map of car B, and (d) represents the feature weight map of the background. By integrating these maps, it becomes apparent that the region corresponding to car A predominantly consists of the feature representation of car A and likewise for car B. And the visible area of the background solely contains the feature representation of the background.

## 3   3D-GOI

In this section, we present the problem definition of 3D-GOI and our three-step inversion method: scene decomposition, coarse estimation, and precise optimization, as depicted in Figure 2.

### 3.1   PROBLEM DEFINITION

The problem we target is similar to the general definition of GAN inversion, with the difference being that we need to invert many more codes than existing methods(1 or 2) as shown in Figure 3. The parameter $w$ in GIRAFFE, which controls the generation of images, can be divided into three categories: object attributes, background attributes, and pose attributes. We use the prefix *obj* to denote object attributes, *bg* for background attributes, and *camera_pose* for pose attributes. As such, $w$ can be denoted as follows:

$$W = \{obj\_shape_i, obj\_app_i, obj\_s_i, obj\_t_i, obj\_r_i,$$
$$bg\_shape, bg\_app, cam\_pose\} \quad i = 1, ..., n, \tag{3}$$

where $obj\_shape$ is the object shape latent code, $obj\_app$ is the object appearance latent code, $obj\_s$ is the object scale code, $obj\_t$ is the object translation code, $obj\_r$ is the object rotation code, $bg\_shape$ is the background shape latent code, $bg\_app$ is the background appearance latent code and $cam\_pose$ is the camera pose matrix. $n$ denotes the $n$ objects. Then, the reconstruction part of the inversion task can be expressed as:

$$W^* = arg \min_W \mathcal{L}(G(W, \theta), I), \tag{4}$$

where $G$ denotes the generator, $\theta$ denotes the parameters of the generator, $I$ is the input image, and $\mathcal{L}$ is the loss function measuring the difference between the generated and input image. According to Equation3, we need to invert a total of $(5n + 3)$ codes. Then, we are able to replace or interpolate any inverted code(s) to achieve multifaceted editing of multiple objects.

### 3.2   SCENE DECOMPOSITION

As mentioned in previous sections, the GIRAFFE generator differs from typical GAN generators in that a large number of codes are involved in generating images, and not a single code controls the generation of all parts of the image. Therefore, it is challenging to transform all codes using just one encoder or optimizer as in typical GAN Inversion methods. A human can easily distinguish each object and some of its features (appearance, shape) from an image, but a machine algorithm requires a large number of high-precision annotated samples to understand what code is expressed at what position in the image.

A straightforward idea is that in images with multiple objects, the attribute codes of an object will map to the corresponding position of the object in the image. For example, translation ($obj\_t$) and rotation ($obj\_r$) codes control the relative position of an object in the scene, scaling ($obj\_s$) and shape ($obj\_shape$) codes determine the contour and shape of the object, and appearance ($obj\_app$) codes control the appearance representation at the position of the object. The image obtained from segmentation precisely encompasses these three types of information, allowing us to invert it and obtain the five attribute codes for the corresponding object. Similarly, for the codes ($bg\_app, bg\_shape$) that

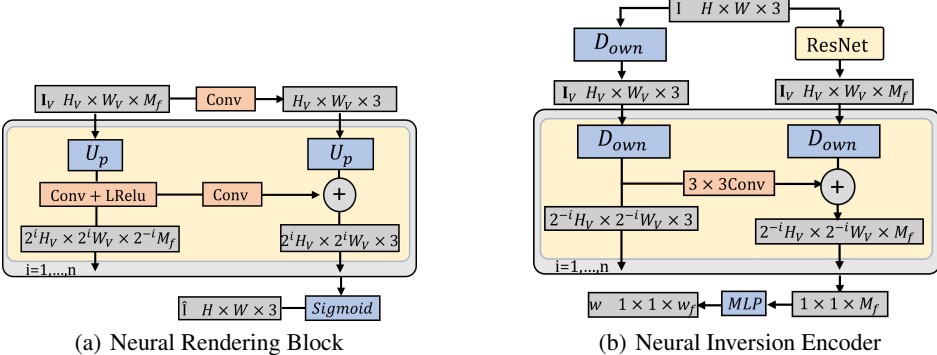

(a) Neural Rendering Block  (b) Neural Inversion Encoder

Figure 5: The design of Neural Inversion Encoder. (a) represents the Neural Rendering Block in GI-RAFFE (Niemeyer & Geiger, 2021), which is an upsampling process to generate image $\hat{I}$. In contrast, (b) illustrates the Neural Inversion Encoder that opposes it, which is a downsampling process. $I$ is the input image, $H, W$ are image height and width. $I_v$ denotes the heatmap of the image, $H_v, W_v$ and $M_f$ are the dimensions of $I_v$, $w$ is the code to be predicted, and $w_f$ is the dimension of $w$. Up means upsampling and Down means downsampling.

generate the background, we can invert them using the segmented image of the background. Note that obtaining $cam\_pose$ requires information from the entire rendered image.

We can qualitatively validate this idea. In Equation 1, we can see that an object's five attribute codes are mapped to the object's feature field and volume density through $h_\theta$. As inferred from Equation 2, the scene's feature field is synthesized by weighting the feature fields of each object by density. Therefore, the reason we see an object appear at its position in the scene is due to its feature field having a high-density weight at the corresponding location. Figure 4 displays the density of different objects at different positions during GIRAFFE's feature field composition process. The redder the color, the higher the density, while the bluer the color, the lower the density. As we discussed, car A exhibits a high-density value within its own area and near-zero density elsewhere - a similar pattern is seen with car B. The background, however, presents a non-uniform density distribution across the entire scene. we can consider that both car A and car B and the background mainly manifest their feature fields within their visible areas. Hence, we apply a straightforward segmentation method to separate each object's feature field and get the codes.

Segmenting each object also has an important advantage: it allows our encoder to pay more attention to each input object or background. As such, we can train the encoder on single-object scenes and then generalize it to multi-object scenes instead of directly training in multi-object scenes that involve more codes, to reduce computation cost.

### 3.3 COARSE ESTIMATION

The previous segmentation step roughly disentangles the codes. Unlike typical encoder-based methods, it's difficult to predict all codes using just one encoder. Therefore, we assign an encoder to each code, allowing each encoder to focus solely on predicting one code. Hence, we need a total of eight encoders. As shown in Figure 2, we input the object segmentation for the object attribute codes ($obj\_shape, obj\_app, obj\_s, obj\_t, obj\_r$), the background segmentation for the background attribute codes ($bg\_shape, bg\_app$), and the original image for pose attribute code ($cam\_pose$). Different objects share the same encoder for the same attribute code.

We allocate an encoder called Neural Inversion Encoder with a similar structure to each code. Neural Inversion Encoder consists of three parts as Figure 5(b) shows. The first part employs a standard feature pyramid over a ResNet (He et al., 2016) backbone like in pSp (Richardson et al., 2021) to extract the image features. The second part, in which we designed a structure opposite to GIRAFFE's Neural rendering Block based on its architecture as Figure 5(a) shows, downsamples the images layer by layer using a Convolutional Neural Network (CNN) and then uses skip connections (He et al., 2016) to combine the layers, yielding a one-dimensional feature. The third layer employs an MLP structure to acquire the corresponding dimension of different codes. Please refer to the Appendix C.1 for the detailed structure of our Neural Inversion Encoder.

---

**Algorithm 1:** Round-robin Optimization

---

**Data:** all codes $w \in W$ predicted by encoders, fixed GIRAFFE generator $G$, input image $I$;

1  Initialize $lr\_w = 10^{-3}, w \in W$ ;

2  **while** *any $lr\_w > 10^{-5}$* **do**

3    **foreach** $w \in W$ **do**

4      Sample $\delta w$;

5      Compute $\delta \mathcal{L}(w)$ using Eq. 6;

6    **end**

7    Compute $rank\_list$ using Eq. 7;

8    **foreach** $w \in rank\_list$ *and $lr\_w > 10^{-5}$* **do**

9      Optimization $w$ with $\mathcal{L}_{opt}$ in Eq. 8 of $I$ and $G(W; \theta)$;

10     **if** *the $\mathcal{L}_{opt}$ ceases to decrease for five consecutive iterations* **then**

11       $lr\_w = lr\_w/2$;

12     **end**

13    **end**

14  **end**

---

Training multiple encoders simultaneously is difficult to converge due to the large number of training parameters. Hence, we use the dataset generated by GIRAFFE for training to retain the true values of each code and train an encoder for one code at a time, to keep the other codes at their true values. Such a strategy greatly ensures smooth training.

During encoder training, we use the Mean Squared Error (MSE) loss, perceptual loss (LPIPS) (Zhang et al., 2018), and identity loss (ID) (He et al., 2020) between the reconstructed image and the original image, to be consistent with most 2D and 3D GAN inversion training methodologies. When training the affine codes (scale $s$, translation $t$, rotation $r$), we find that different combinations of values produce very similar images, e.g., moving an object forward and increasing its scale yield similar results. However, the encoder can only predict one value at a time, hence we add the MSE loss of the predicted $s,t,r$ values, and their true values, to compel the encoder to predict the true value.

$$\mathcal{L}_{enc} = \lambda_1 L_2 + \lambda_2 L_{lpips} + \lambda_3 L_{id}, \tag{5}$$

where $\lambda_i, i = 1, 2, 3$ represent the ratio coefficient between various losses. When training $obj\_s, obj\_t, obj\_r$ code, the $L_2$ loss includes the MSE loss between the real values of $obj\_s, obj\_t, obj\_r$ and their predicted values.

### 3.4 PRECISE OPTIMIZATION

Next, we optimize the coarse codes predicted by the encoder. Through experiments, we have found that using a single optimizer to simultaneously optimize all latent codes tends to converge to local minima. To circumvent this, we employ multiple optimizers, each handling a single code as in the coarse estimation. The optimization order plays a crucial role in the overall outcome due to the variance of the disparity between the predicted and actual values across different encoders, and the different impact of code changes on the image, e.g., changes to $bg\_shape$ and $bg\_app$ codes controlling background generation mostly would have a larger impact on overall pixel values. Prioritizing the optimization of codes with significant disparity and a high potential for changing pixel values tends to yield superior results in our empirical experiments. Hence, we propose an automated round-robin optimization algorithm (Algorithm 1) to sequentially optimize each code based on the image reconstructed in each round.

Algorithm 1 aims to add multiple minor disturbances to each code, and calculate the loss between the images reconstructed before and after the disturbance and the original image. A loss increase indicates that the current code value is relatively accurate, hence its optimization order can be put later. A loss decrease indicates that the current code value is inaccurate and thus should be prioritized. For multiple codes that demand prioritized optimization, we compute their priorities using the partial derivatives of the loss variation and perturbation. We do not use backpropagation automatic

differentiation here to ensure the current code value remains unchanged.

$$\delta\mathcal{L}(w) = \mathcal{L}(G(W - \{w\}, w + \delta w, \theta), I) - \mathcal{L}(G(W, \theta), I), \tag{6}$$

$$rank\_list = F_{rank}(\delta\mathcal{L}(w), \frac{\delta\mathcal{L}(w)}{\delta w}), \tag{7}$$

where $w \in W$ is one of the codes and $\delta w$ represents the minor disturbance of $w$. For the rotation angle $r$, we have found that adding a depth loss can accelerate its optimization. Therefore, the loss $\mathcal{L}$ during the optimization stage can be expressed as:

$$\mathcal{L}_{opt} = \lambda_1 L_2 + \lambda_2 L_{lpips} + \lambda_3 L_{id} + \lambda_4 L_{deep}. \tag{8}$$

This optimization method allows for more precise tuning of the codes for more accurate reconstruction and editing of the images.

## 4 EXPERIMENT

**Datasets.** To obtain the true values of the 3D information in GIRAFFE for stable training performance, we use the pre-trained model of GIRAFFE on the CompCars (Yang & Li, 2015) dataset and Clevr (Johnson et al., 2017) dataset to generate training datasets. For testing datasets, we also use GIRAFFE to generate images for multi-car datasets denoted as *G-CompCars* (CompCars is a single car image dataset) and use the original Clevr dataset for multi-geometry dataset (Clevr is a dataset that can be simulated to generate images of multiple geometries). We follow the codes setup in GIRAFFE. For CompCars, we use all the codes from Equation 3. For Clevr, we fixed the rotation, scale, and camera pose codes of the objects. For experiments on facial data, we utilized the FFHQ (Karras et al., 2019) dataset for training and the CelebA-HQ (Karras et al., 2017) dataset for testing.

**Baselines.** In the comparative experiments for our Neural Inversion Encoder, we benchmarked encoder-based inversion methods such as e4e (Tov et al., 2021) and pSp (Richardson et al., 2021), which use the 2D GAN StyleGAN2 (Karras et al., 2020) as the generator, and E3DGE (Lan et al., 2023) and TriplaneNet (Bhattarai et al., 2023) that employ the 3D GAN EG3D (Chan et al., 2022) as the generator, on the generator of GIRAFFE. Additionally, we compared our encoder on StyleGAN2 with SOTA inversion methods HyperStyle (Alaluf et al., 2022) and HFGI (Wang et al., 2022) for StyleGAN2.

**Metrics.** We use Mean Squared Error (MSE), perceptual similarity loss (LPIPS) (Zhang et al., 2018), and identity similarity (ID) to measure the quality of image reconstruction.

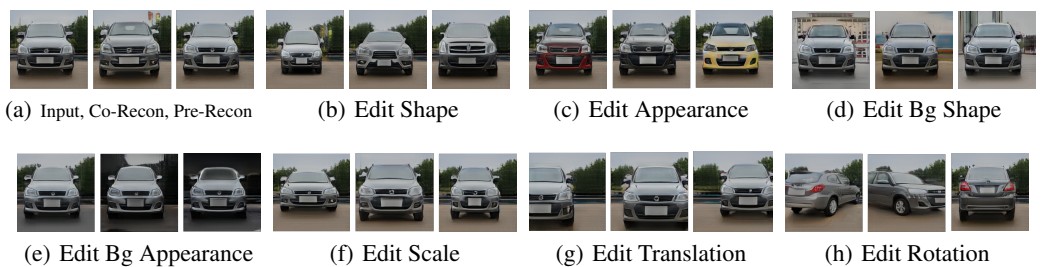

(a) Input, Co-Recon, Pre-Recon     (b) Edit Shape     (c) Edit Appearance     (d) Edit Bg Shape

(e) Edit Bg Appearance     (f) Edit Scale     (g) Edit Translation     (h) Edit Rotation

Figure 6: Single-object editing on *G-CompCars* dataset. Co-Recon: coarse reconstruction. Pre-Recon: precise reconstruction.

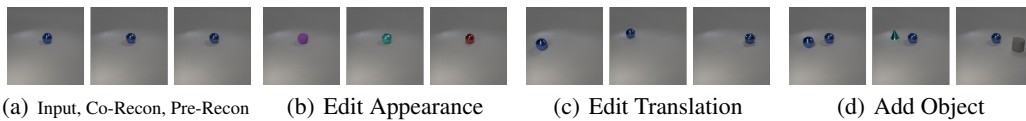

(a) Input, Co-Recon, Pre-Recon     (b) Edit Appearance     (c) Edit Translation     (d) Add Object

Figure 7: Single-object editing on *Clevr* dataset.

### 4.1 3D GAN OMNI-INVERSION

#### 4.1.1 SINGLE-OBJECT MULTIFACETED EDITING

In Figure 6 and Figure 7, (a) depict the original images, the coarsely reconstructed images produced by the Neural Inversion Encoder, and the precisely reconstructed images obtained via round-robin optimization. As Figure 7 shows, the simple scene structure of the Clevr dataset allows us to achieve remarkably accurate results using only the encoder (Co-Recon). However, for car images in Figure 6, predicting precise codes using the encoder only becomes challenging, necessitating the employment of the round-robin optimization algorithm to refine the code values for precise reconstruction (Pre-Recon). Figure 6 (b)-(h) and Figure 7 (b)-(d) show the editing results for different codes. As noted in Section 3.3, moving an object forward and increasing its scale yield similar results. Due to space constraints, please refer to the Appendix D.1 for more results like camera pose and shape editing.

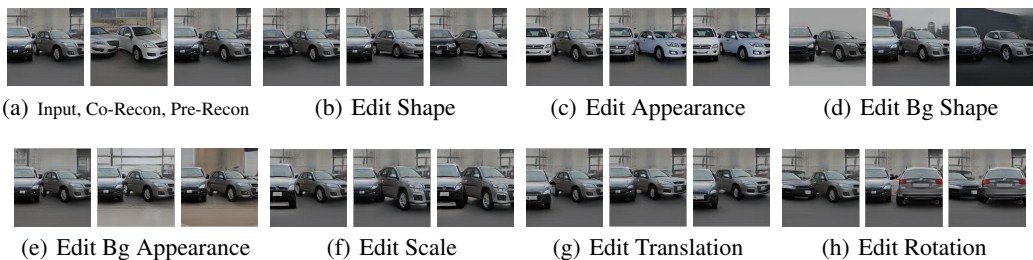

| (a) Input, Co-Recon, Pre-Recon | (b) Edit Shape | (c) Edit Appearance | (d) Edit Bg Shape |

| (e) Edit Bg Appearance | (f) Edit Scale | (g) Edit Translation | (h) Edit Rotation |

Figure 8: Multi-object editing on *G-CompCars* dataset.

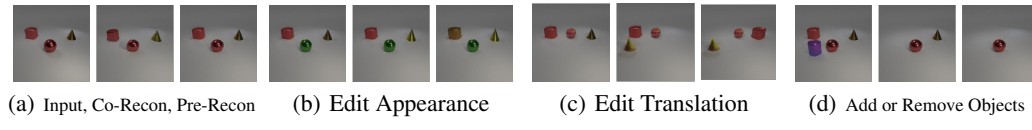

| (a) Input, Co-Recon, Pre-Recon | (b) Edit Appearance | (c) Edit Translation | (d) Add or Remove Objects |

Figure 9: Multi-object editing on *Clevr* dataset.

#### 4.1.2 MULTI-OBJECT MULTIFACETED EDITING

We notice that the prediction for some object parameters ($obj\_shape$, $obj\_app$, $obj\_s$, $obj\_t$) are quite accurate. However, the prediction for the background codes deviates significantly. We speculate this is due to the significant differences in segmentation image input to the background encoder between multi-object scenes and single-object scenes. Therefore, background reconstruction requires further optimization. Figure 8 and Figure 9 depict the multifaceted editing outcomes for two cars and multiple Clevr objects, respectively. The images show individual edits of two objects in the left and middle images and collective edits at the right images in Figure 8 (b-c) and (f-h). As demonstrated in Figure 8, the predictive discrepancy between the background and the rotation angle of the car on the left is considerable, requiring adjustments through the round-robin optimization algorithm. As illustrated in Figure 1, 2D/3D GAN inversion methods can not inverse multi-object scenes. More images pertaining to multi-object editing can be found in the Appendix D.2.

### 4.2 COMPARISON EXPERIMENT OF NEURAL INVERSION ENCODER

For fair comparison and to eliminate the impact of the generator on the quality of the inverted image generation, we trained the encoders from the baseline methods by connecting them to the GIRAFFE generator using our Neural Inversion Encoder training approach and compared them with our Neural Inversion Encoder. At the same time, we also connected our encoder to StyleGAN2 and compared it with inversion methods based on StyleGAN2, thereby demonstrating the efficiency of our encoder design. Table 1 quantitatively displays the comparison results on both the GIRAFFE and StyleGAN2 generators. The results show that our Neural Inversion Encoder consistently outperforms baseline methods. Please refer to the qualitative results on the images in the Appendix D.3.

Table 1: Reconstruction quality of different GAN inversion encoders using the generator of GIRAFFE and StyleGAN2. ↓ indicates the lower the better and ↑ indicates the higher the better.

| Method | GIRAFFE for Generator | | | StyleGAN2 for Generator | | |
|---|---|---|---|---|---|---|
| | MSE ↓ | LPIPS ↓ | ID↑ | MSE ↓ | LPIPS ↓ | ID↑ |
| e4e (Tov et al., 2021) | 0.031 | 0.306 | 0.867 | 0.052 | 0.200 | 0.502 |
| pSp (Richardson et al., 2021) | 0.031 | 0.301 | 0.877 | 0.034 | 0.172 | 0.561 |
| HyperStyle (Alaluf et al., 2022) | - | - | - | 0.019 | 0.091 | 0.766 |
| HFGI (Wang et al., 2022) | - | - | - | 0.023 | 0.124 | 0.705 |
| TriplaneNet (Bhattarai et al., 2023) | 0.029 | 0.296 | 0.870 | - | - | - |
| E3DGE (Lan et al., 2023) | 0.031 | 0.299 | 0.881 | - | - | - |
| 3D-GOI(Ours) | **0.024** | **0.262** | **0.897** | **0.017** | **0.098** | **0.769** |

Table 2: Ablation Study of the Neural Inversion Encoder.

| Method | MSE ↓ | LPIPS↓ | ID ↑ |
|---|---|---|---|
| w/o NIB | 0.023 | 0.288 | 0.856 |
| w/o MLP | 0.015 | 0.183 | 0.878 |
| 3D-GOI | **0.010** | **0.141** | **0.906** |

Table 3: The quantitative metrics of ablation study of the Round-robin Optimization algorithm.

| Method | MSE ↓ | LPIPS ↓ | ID↑ |
|---|---|---|---|
| Order1 | 0.016 | 0.184 | 0.923 |
| Order2 | 0.019 | 0.229 | 0.913 |
| Order3 | 0.019 | 0.221 | 0.911 |
| 3D-GOI | **0.008** | **0.128** | **0.938** |

## 4.3 ABLATION STUDY

We conducted ablation experiments separately for the proposed Neural Inversion Encoder and the Round-robin Optimization algorithm.

Table 2 displays the average ablation results of the Neural Inversion Encoder on various attribute codes, where NIB refers to Neural Inversion Block (the second part of the encoder) and MLP is the final part of the encoder. The results clearly show that our encoder structure is extremely effective and can predict code values more accurately. Please find the complete results in the Appendix D.5.

For the Round-robin optimization algorithm, we compared it with three fixed optimization order algorithms on both single-object and multi-object scenarios. The three fixed sequences are as follows:

$$Order1 : bg\_shape, bg\_app, \{obj\_r_i, obj\_t_i, obj\_s_i\}_{i=1}^{N}, \{obj\_shape_i, obj\_app_i\}_{i=1}^{N}, camera\_pose$$

$$Order2 : \{obj\_r_i, obj\_t_i, obj\_s_i\}_{i=1}^{N}, \{obj\_shape_i, obj\_app_i\}_{i=1}^{N}, bg\_shape, bg\_app, camera\_pose$$

$$Order3 : camera\_pose, \{obj\_shape_i, obj\_app_i\}_{i=1}^{N}, \{obj\_r_i, obj\_t_i, obj\_s_i\}_{i=1}^{N}, bg\_shape, bg\_app$$

$\{\}_{i=1}^{N}$ indicates that the elements inside $\{\}$ are arranged in sequence from 1 to N. There are many possible sequence combinations, and here we chose **the three with the best results** for demonstration. Table 3 is the quantitative comparison of the four methods. As shown, our method achieves the best results on all metrics, demonstrating the effectiveness of our Round-robin optimization algorithm. As mentioned in 3.4, optimizing features like the image background first can enhance the optimization results. Hence, Order1 performs much better than Order2 and Order3. Please see the Appendix D.5 for qualitative comparisons of these four methods on images.

## 5 CONCLUSION

This paper introduces a 3D GAN inversion method, 3D-GOI, that enables multifaceted editing of scenes containing multiple objects. By using a segmentation approach to separate objects and background, then carrying out a coarse estimation followed by a precise optimization, 3D-GOI can accurately obtain the codes of the image. These codes are then used for multifaceted editing. To the best of our knowledge, 3D-GOI is the first method to attempt multi-object & multifaceted editing. We anticipate that 3D-GOI holds immense potential for future applications in fields such as VR/AR, and the Metaverse.

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

## A    RELATED WORK

**2D/3D GANs.**    2D GAN maps a distribution from the latent space to the image space. It generally consists of two parts: a generator and a discriminator. The generated image needs to deceive the discriminator, while the discriminator needs to discern whether the image was generated. Through this adversarial strategy, the images produced by the trained generator gradually approximate the distribution of the real image dataset. There are many variants of 2D GANs. For example, Big-GAN (Brock et al., 2018) increases the batch size and uses a simple truncation trick to finely control the trade-off between sample fidelity and variety. CycleGAN (Zhu et al., 2017) feeds an input image into the generator to get a result, then inputs it back into the generator, and by minimizing the consistency loss between the input and its result, it achieves style transfer. StyleGAN (Karras et al., 2019) maps a latent code into multiple style codes, allowing for detailed style control of images.

3D GANs usually combine 2D GANs with some form of 3D representation, such as NeRF (Mildenhall et al., 2021), which have demonstrated excellent abilities to generate complex scenes with multi-view consistency. Broadly, 3D GANs can be classified into two categories: explicit and implicit models. Explicit models like HoloGAN (Nguyen-Phuoc et al., 2019) enable explicit control over the pose of the resulting object through rigid body transformations of the learned 3D features. BlockGAN (Nguyen-Phuoc et al., 2020) generates foreground and background 3D features separately, combining them into a complete 3D scene representation that is ultimately rendered into a realistic image. On the other hand, implicit models generally perform better. Many of these models take inspiration from NeRF (Mildenhall et al., 2021), representing images as neural radiance fields and using volume rendering to generate photorealistic images in a continuous view. EG3D (Chan et al., 2022) introduces an explicit-implicit hybrid network architecture that produces high-quality 3D geometries. GRAF (Schwarz et al., 2020) integrates shape and appearance coding within the generation process, which facilitates independent manipulation of the shape and appearance of the generated vehicle and furniture images. Moreover, the presence of 3D information provides additional control over the camera pose, contributing to the flexibility of the generated outputs. GI-RAFFE (Niemeyer & Geiger, 2021) extends GRAF to multi-object scenes by considering image as the composition of multiple objects in the foreground through affine transformation and the background rendered at a specific camera viewpoint. In this work, we select GIRAFFE as the 3D GAN model to be inverted.

**2D/3D GAN Inversion.**    GAN inversion is the opposite process of GANs, obtaining the latent code of an input image under a certain generator and modifying the latent code to perform image editing operations. Current 2D GAN inversion methods can be divided into optimization-based, encoder-based, and hybrid methods. Optimization-based methods  (Abdal et al., 2019; Zhu et al., 2016; Huh et al., 2020) directly optimize the initial code, requiring very accurate initial values. Encoder-based methods  (Perarnau et al., 2016; Richardson et al., 2021; Wei et al., 2022) can map images directly to latent code but generally cannot achieve full reconstruction. Hybrid-based methods (Zhu et al., 2020; Bau et al., 2019) combine these two approaches: they first employ an encoder to map the image to a suitable latent code, and then perform optimization. Currently, most 2D GANs only have one latent code to generate an image [1]. Therefore, the 2D GAN inversion task can be represented as:

$$\omega^* = arg \min_{\omega} \mathcal{L}(G(\omega, \theta), I), \tag{9}$$

where $\omega$ is the latent component, $G$ denotes the generator, $\theta$ denotes the parameters of the generator, $I$ is the input image, and $\mathcal{L}$ is the loss function measuring the difference between the generated and input image.

Typically, 3D GANs have an additional camera pose parameter compared to 2D GANs, making it more challenging to obtain latent codes during inversion. Current methods like SPI  (Yin et al., 2022) use a symmetric prior for faces to generate images with different perspectives, while  (Ko et al., 2023) employs a pre-trained estimator to achieve better initialization and utilizes pixel-level depth calculated from the NeRF parameters for improved image reconstruction.

---

[1]Although StyleGAN can be controlled by multiple style codes, these codes are all generated from a single initial latent code, indicating their interrelations. Hence only one encoder is needed to predict all the codes during inversion.

Currently, there are only limited works on 3D GAN inversion (Xie et al., 2022; Deng et al., 2022; Lan et al., 2022) which primarily focus on creating novel perspectives of human faces using specialized face datasets considering generally only two codes: camera pose code and the latent code. Hence its inversion task can be represented as:

$$\boldsymbol{\omega}^*, \boldsymbol{c}^* = arg \min_{\boldsymbol{\omega}, \boldsymbol{c}} \mathcal{L}(G(\boldsymbol{\omega}, \boldsymbol{c}, \theta), I). \tag{10}$$

A major advancement of 3D-GOI is the capability to invert more independent codes compared with other inversion methods, as Figure 3 shows, in order to perform multifaceted edits on multi-object images.

## B PRELIMINARY

**NeRF** (Mildenhall et al., 2021) is a recently rising approach for 3D reconstruction tasks that employs a neural radiance field to represent a scene. It allows for mapping high-dimensional positional codes from any viewing direction $\boldsymbol{d}$ and spatial coordinates $\boldsymbol{x}$ to color $\boldsymbol{c}$ and opacity values $\sigma$ and then synthesizes images corresponding to the specified view using a volume rendering equation. We use Equation 11 to succinctly describe this process:

$$(\gamma(\boldsymbol{x}), \gamma(\boldsymbol{d})) \xrightarrow{f_\theta} (\sigma, \boldsymbol{c})$$
$$\mathbb{R}^{L_x} \times \mathbb{R}^{L_d} \xrightarrow{f_\theta} \mathbb{R}^+ \times \mathbb{R}^3 \tag{11}$$

where $\gamma$ represents the positional encoding function utilized to incorporate high-dimensional information into $\boldsymbol{x}$ and $\boldsymbol{d}$ and obtained the output $\gamma(\mathbf{x})$, $\gamma(\boldsymbol{d})$ of dimension $L_x, L_d$, respectively. $\gamma$ is typically represented using trigonometric functions, such as $\gamma(t, L) = (sin(2^0 t\pi), cos(2^0 t\pi), ..., sin(2^{L-1} t\pi), cos(2^{L-1} t\pi))$. $\theta$ represents the parameters of the mapping function $f$.

Equation 12 delineates the volume rendering formula that predicts color $C(\boldsymbol{r})$ for a camera ray $\boldsymbol{r}(t) = \boldsymbol{o} + t\boldsymbol{d}$ within the near and far bounds $tn$ and $tf$. Here, $T(t)$ signifies the cumulative transmittance along the ray from $tn$ to $t$.

$$C(\mathbf{r}) = \int_{t_n}^{t_f} T(t)\sigma(r(t))c(r(t), d)dt,$$
$$where \quad T(t) = exp(-\int_{t_n}^{t} \sigma(r(s))ds) \tag{12}$$

**GRAF** (Schwarz et al., 2020) is a generative neural radiance field adding additional latent codes like object shape $\boldsymbol{z_s}$ and appearance $\boldsymbol{z_a}$ to NeRF, allowing control not only the shape and appearance of the object but also the camera pose of the image. $\boldsymbol{z_s}, \boldsymbol{z_a} \sim \mathcal{N}(0, I)$ and the mapping function $g_\theta$ of the radiance field of GRAF can be expressed as follows:

$$(\gamma(\boldsymbol{x}), \gamma(\boldsymbol{d}), \boldsymbol{z_s}, \boldsymbol{z_a}) \xrightarrow{g_\theta} (\sigma, \boldsymbol{c})$$
$$\mathbb{R}^{L_x} \times \mathbb{R}^{L_d} \times \mathbb{R}^{M_s} \times \mathbb{R}^{M_a} \xrightarrow{g_\theta} \mathbb{R}^+ \times \mathbb{R}^3, \tag{13}$$

where $Ms$ and $Ma$ are the dimensions of $z_s$ and $z_a$, respectively. GRAF renders images using a volume rendering formula similar to that of NeRF.

**GIRAFFE** (Niemeyer & Geiger, 2021) perceives an image scene as a composition of the background and multiple foreground objects, each subjected to affine transformations. Each object can be manipulated and placed at a specific location $k(\boldsymbol{x})$ in the image through operations of scaling $\boldsymbol{S}$, translation $\boldsymbol{t}$, and rotation $\boldsymbol{R}$:

$$k(\boldsymbol{x}) = \boldsymbol{R} \cdot \boldsymbol{S} \cdot \boldsymbol{x} + \boldsymbol{t}, \tag{14}$$

where $\mathbf{x}$ is the spatial coordinate in the object space.

To better compose scenes, GIRAFFE replaces the three-dimensional color output in GRAF's Equation 13 with a high-dimensional feature field. GIRAFFE renders in scene space and evaluates the

feature field in the object space. Hence, the mapping function of radiance field $h_\theta$ of GIRAFFE in object space can be expressed as follows:

$$(\gamma(k^{-1}(\boldsymbol{x})), \gamma(k^{-1}(\mathbf{d})), \boldsymbol{z_s}, \boldsymbol{z_a}) \xrightarrow{h_\theta} (\sigma, \boldsymbol{f})$$
$$\mathbb{R}^{L_x} \times \mathbb{R}^{L_d} \times \mathbb{R}^{M_s} \times \mathbb{R}^{M_a} \xrightarrow{h_\theta} \mathbb{R}^+ \times \mathbb{R}^{M_f}, \quad (15)$$

where $k^{-1}$ is the inverse function of $k$, $M_f$ is the dimension of the feature field $\boldsymbol{f}$.

In the construction of multi-object scenes, GIRAFFE employs a compositing operation $C$ to merge the feature fields of multiple objects and the background together. The features at $(\boldsymbol{x}, \boldsymbol{d})$ can be expressed as:

$$C(\boldsymbol{x}, \boldsymbol{d}) = (\sigma, \frac{1}{\sigma} \sum_{i=1}^{N} \sigma_i \boldsymbol{f_i}), where \quad \sigma = \sum_{i=1}^{N} \sigma_i, \quad (16)$$

where N is the number of objects plus one (the background), $\sigma_i$ and $\boldsymbol{f_i}$ represent the density value and feature field of the $i - th$ object (or the background).

The rendering process of GIRAFFE can be divided into two stages. In the first stage, feature fields are used instead of color for volume rendering like in NeRF to get a low-resolution feature map:

$$\boldsymbol{f} = \sum_{i=1}^{N_s} \tau_j \alpha_j \boldsymbol{f_j}, \quad \tau_j = \prod_{k=1}^{j-1} (1 - \alpha_k), \quad \alpha_j = 1 - e^{-\sigma_j \delta_j}, \quad (17)$$

where $\alpha_j$ is the alpha value of the coordinates $\boldsymbol{x_j}$, $\tau_j$ represents the transmittance, and $\delta_j = ||\boldsymbol{x_{j+1}} - \boldsymbol{x_j}||_2$ is the distance between the neighboring sampled points $\boldsymbol{x_{j+1}}$ and $\boldsymbol{x_j}$. The second stage is called neural rendering, which transforms low-resolution feature maps into high-resolution images through an upsampling network.

## C    IMPLEMENTATION

### C.1    NEURAL INVERSION ENCODER

The first part of our encoder uses ResNet50 to extract features. In the second part, we downsample the extracted features (512-dimensional) and the input RGB image (3-dimensional) together. The two features are added together through skip connections, as shown in Figure 5. In the downsampling module, we use a 2D convolution with a kernel of 3 and a stride of 1, and the LeakyReLU activation function, to obtain a 256-dimensional intermediate feature. For object shape/appearance attributes, the output dimension is 256, and we use four Fully Connected Layers $\{4 \times FCL(256, 256)\}$ to get the codes. For background shape/appearance attributes, the output dimension is 128, we use $\{FCL(256, 128) + 3 \times FCL(128, 128)\}$ to get the codes. For object scale/translation attributes, the output dimension is 3, and we use the network $\{FCL(2^i, 2^{i-1}) + FCL(8, 3), i = 8, .., 4\}$ to get the codes. For camera pose and rotation attributes, the output dimension is 1, and we use a similar network $\{FCL(2^i, 2^{i-1}) + FCL(8, 1), i = 8, .., 4\}$ to get the codes.

### C.2    TRAINING AND OPTIMIZATION PROCESS

Our training and optimization are carried out on a single NVIDIA A100 SXM GPU with 40GB of memory, using the Adam optimizer. The initial learning rate is set to $10^{-4}$ and $10^{-3}$ for training and optimization, respectively. Encoder training employs a batch size of 50. Each encoder took about 12 hours to train, and optimizing a single image of a complex multi-object scene took about 1 minute. For rotation features, it is difficult for the encoder to make accurate predictions for some images. Therefore, in our experiments, we uniformly sample 20 values in the range of [0, 360°] for the rotation parameters with large deviations. We select the value that minimizes the loss in Equation 5 as the initial value for the optimization stage.

For LPIPS loss (Zhang et al., 2018), we employ a pre-trained AlexNet (Krizhevsky et al., 2017). For ID calculation, we employ a pre-trained Arcface (Deng et al., 2019) model in human face datasets and employ a pre-trained ResNet-50 (Russakovsky et al., 2015) model in the car dataset. For depth

Table 4: Architecture comparison for different GAN inversion methods. SG2 indicates StyleGAN2. "2D/3D" indicates whether 2D or 3D editing is possible. "object" indicates whether the method can edit a single object or multiple objects. "code" indicates the number of codes that the method can invert.

| Method | Generator | 2D/3D | object | code |
|---|---|---|---|---|
| e4e (Tov et al., 2021) | SG2 | 2D | single | 1 |
| pSp (Richardson et al., 2021) | SG2 | 2D | single | 1 |
| PTI (Roich et al., 2022) | SG2 | 2D | single | 1 |
| HyperStyle (Alaluf et al., 2022) | SG2 | 2D | single | 1 |
| HFGI (Wang et al., 2022) | SG2 | 2D | single | 1 |
| TriPlaneNet (Bhattarai et al., 2023) | EG3D | 3D | single | 2 |
| E3DGE (Lan et al., 2023) | EG3D | 3D | single | 2 |
| SPI (Yin et al., 2022) | EG3D | 3D | single | 2 |
| 3D-GOI | GIRAFFE | 2D/3D | single/multi | 5n+3 |

loss, we use the pre-trained Dense Prediction Transformer model. We set $\lambda_1 = 1$, $\lambda_2 = 0.8$, and $\lambda_3 = 0.2$ in Equation 5, and in Equation 8 the first three $\lambda$ parameters remain the same and $\lambda_4 = 1$.

The round-robin optimization algorithm works well when the discrepancy between the coarse estimation of the Neural Inversion Encoder and the actual results is not too large. This is because in the presence of a slight perturbation in the codes, an increase in the loss of Equation 6 doesn't necessarily conclude that the code has reached its true value. Otherwise, if the encoder cannot make a rough prediction of the code, or if one wishes to forgo using the encoder and rely solely on the optimization method, we offer a program for manually selecting the current optimization code interactively. This allows the image to be manually optimized to a certain degree of difference from the original image before using the round-robin optimization algorithm for automatic optimization.

# D ADDITIONAL RESULTS

**Baselines.** We added another 2D GAN inversion method based on StyleGAN2 called PTI (Roich et al., 2022), and a 3D GAN inversion method based on EG3D named SPI (Yin et al., 2022), to validate the performance of our method in the novel viewpoint synthesis task. Table 4 compares the structures and capabilities of various GAN Inversion methods.

## D.1 SINGLE-OBJECT MULTIFACETED EDITING

Figure 10 and 11 depict the additional results of our multifaceted edits on a single object.

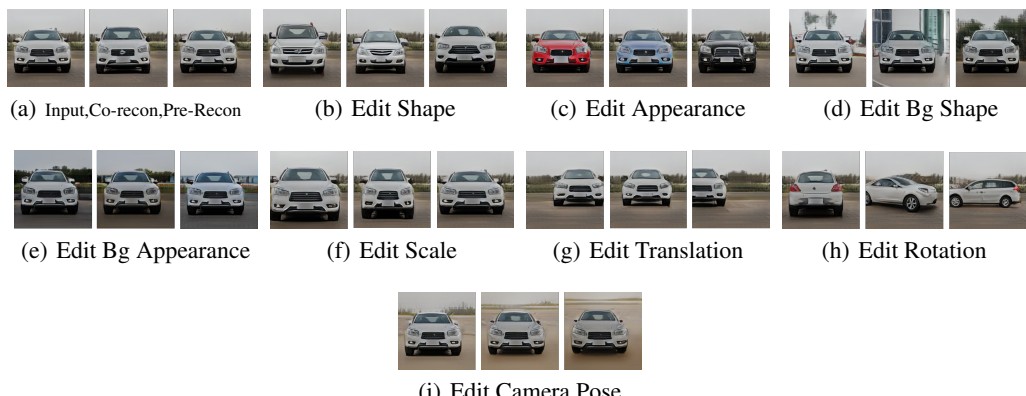

(a) Input,Co-recon,Pre-Recon    (b) Edit Shape    (c) Edit Appearance    (d) Edit Bg Shape

(e) Edit Bg Appearance    (f) Edit Scale    (g) Edit Translation    (h) Edit Rotation

(i) Edit Camera Pose

Figure 10: Single-object editing performance on G-CompCars dataset.

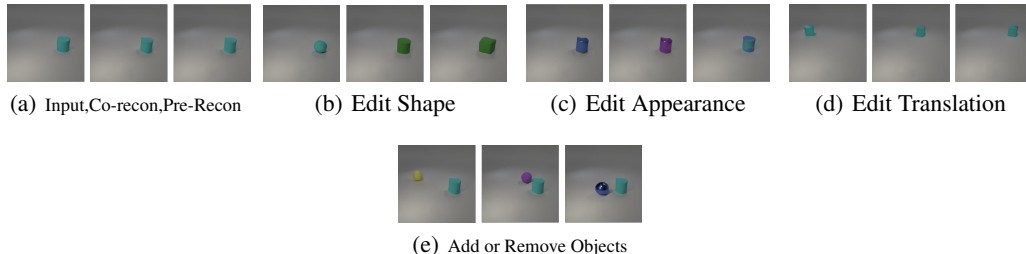

Figure 11: Single-object editing performance on Clevr dataset.

## D.2 MULTI-OBJECT MULTIFACETED EDITING

As shown in the Figure 12 and 13, we demonstrate the additional results of our multifaceted edits on multiple objects.

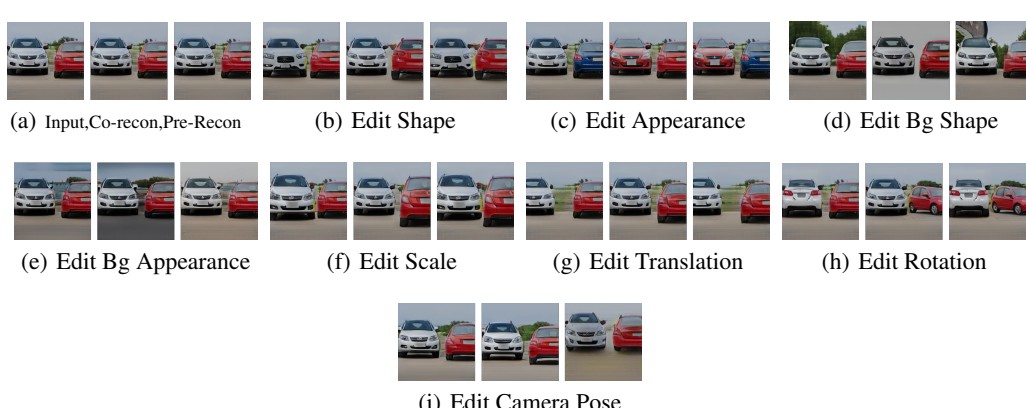

Figure 12: Multi-object editing performance on G-CompCars dataset.

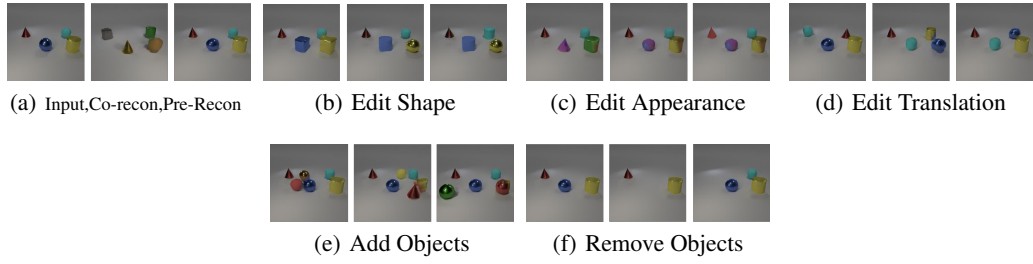

Figure 13: Multi-object editing performance on Clevr dataset.

## D.3 COMPARISON EXPERIMENT OF NEURAL INVERSION ENCODER

Figure 14 shows the performance comparison between our Neural Inversion Encoder and other baseline encoders using the GIRAFFE generator under the same training settings. Evidently, our method achieves the best results in both single-object and multi-object inversion reconstructions.

Figure 15 shows the performance comparison between our method and the baselines using Style-GAN2 as the generator. Our method clearly outperforms the baselines in the inversion of details such as hair and teeth.

As such, we can conclude that our Neural Inversion Encoder performs excellent inversion on different 2D StyleGAN2 and 3D GIRAFFE, both qualitatively and quantitatively.

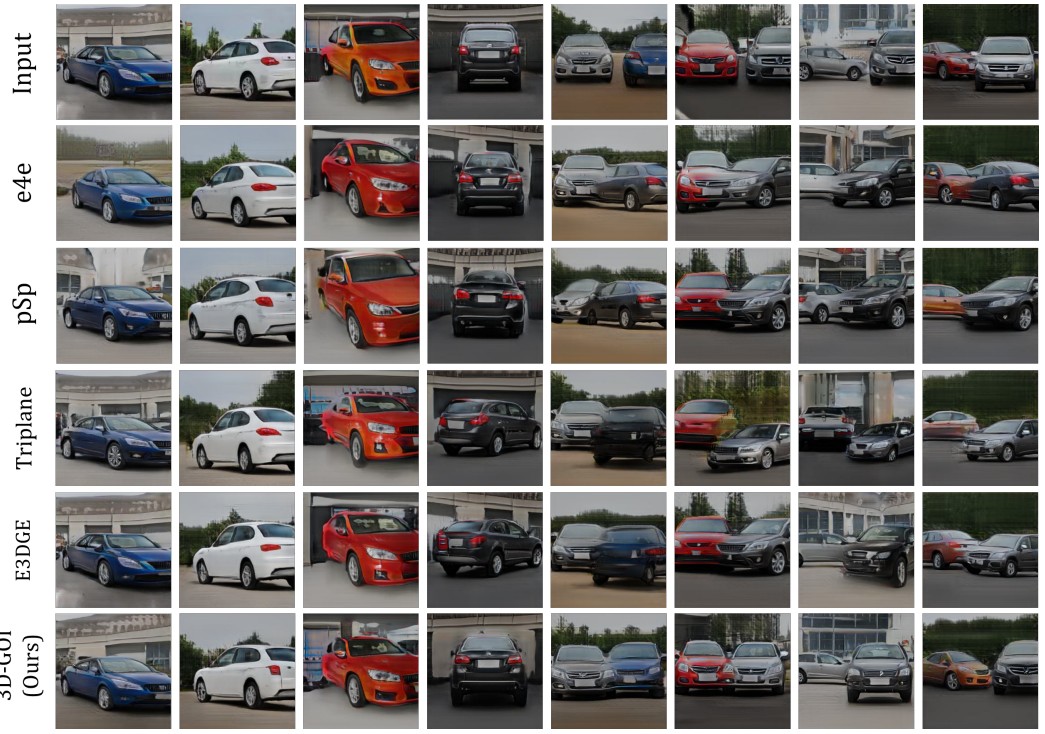

Figure 14: Reconstruction results of different GAN inversion encoders using the generator of GIRAFFE.

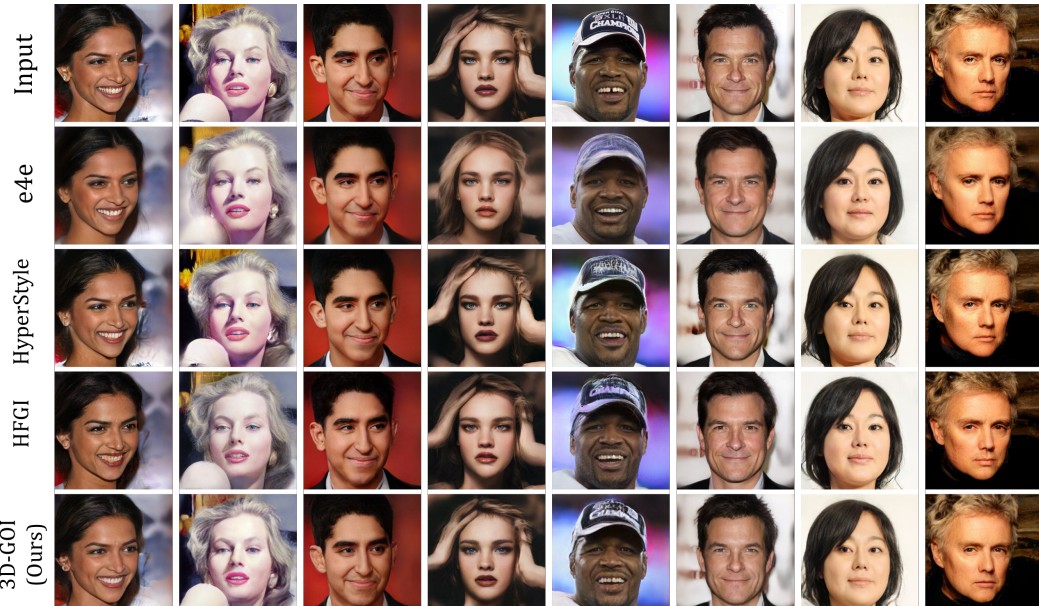

Figure 15: Reconstruction results of different GAN inversion encoders using the generator of StyleGAN2.

### D.4    NOVEL VIEWS SYNTHESIS FOR HUMAN FACES

We also test the synthesis of novel views of the face, which is a minor ability of 3D-GOI yet the major ability of existing 3D GAN inversion methods. Figure 16 shows that our method has better performance than the latest 3D inversion method SPI (Yin et al., 2022) and some advanced 2D inversion methods that can generate novel views such as PTI (Roich et al., 2022) and SG2(StyleGAN2) (Karras et al., 2020).

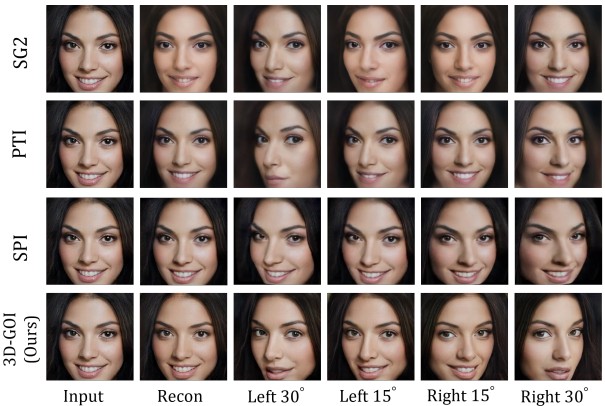

Figure 16: Novel views synthesis for human faces of different GAN inversion methods.

Table 5: Ablation Study of the Neural Inversion Encoder of different attribute codes.

| Method | attribute codes | MSE ↓ | LPIPS↓ | ID ↑ |
|---|---|---|---|---|
| 3D-GOI (w/o NIB) | $obj\_shape$ | 0.046 | 0.412 | 0.811 |
| | $obj\_app$ | 0.006 | 0.092 | 0.907 |
| | $obj\_s$ | 0.025 | 0.269 | 0.856 |
| | $obj\_t$ | 0.036 | 0.340 | 0.848 |
| | $obj\_r$ | 0.031 | 0.343 | 0.805 |
| | $bg\_shape$ | 0.030 | 0.400 | 0.812 |
| | $bg\_app$ | 0.009 | 0.155 | 0.881 |
| | $cam\_pose$ | 0.001 | 0.289 | 0.929 |
| | average | 0.023 | 0.288 | 0.856 |
| 3D-GOI (w/o MLP) | $obj\_shape$ | 0.030 | 0.286 | 0.850 |
| | $obj\_app$ | 0.004 | 0.075 | 0.916 |
| | $obj\_s$ | 0.012 | 0.157 | 0.889 |
| | $obj\_t$ | 0.016 | 0.199 | 0.877 |
| | $obj\_r$ | 0.025 | 0.280 | 0.827 |
| | $bg\_shape$ | 0.022 | 0.316 | 0.837 |
| | $bg\_app$ | 0.006 | 0.120 | 0.898 |
| | $cam\_pose$ | 0.001 | 0.029 | 0.929 |
| | average | 0.015 | 0.183 | 0.878 |
| 3D-GOI | $obj\_shape$ | 0.008 | 0.116 | 0.913 |
| | $obj\_app$ | 0.005 | 0.084 | 0.931 |
| | $obj\_s$ | 0.005 | 0.084 | 0.924 |
| | $obj\_t$ | 0.010 | 0.138 | 0.905 |
| | $obj\_r$ | 0.022 | 0.257 | 0.855 |
| | $bg\_shape$ | 0.021 | 0.332 | 0.853 |
| | $bg\_app$ | 0.005 | 0.116 | 0.922 |
| | $cam\_pose$ | 0.001 | 0.002 | 0.941 |
| | average | 0.010 | 0.141 | 0.906 |

## D.5 ABLATION STUDY

Table 5 shows the results of the ablation experiments on each attribute encoder. It shows that our added NIB structure can greatly improve the prediction accuracy, and that $obj/bg\_shape$ and rotation are more difficult to predict than other codes.

Figure 17 shows the result of using only one optimizer for all codes. For a single object image, even though our encoder can estimate the codes more accurately as shown in Figure 14, the optimizer is still unable to reconstruct the image accurately, which is even more obvious for multi-object codes that require more codes to be controlled.



Figure 17: The result of optimizing all codes using only one optimizer.

Table 6: The comparison of encoder-based 3D inversion methods for computational costs.

| Method | parameter numbers | FLOPs | time(s) |
|---|---|---|---|
| E3DGE (single encoder) | 90M | 50G | 0.07 |
| TriplaneNet (single encoder) | 247M | 112G | 0.11 |
| 3D-GOI(multi encoders) | 169M | 165G | 0.08 |

Figure 18 is a qualitative comparison of the four methods. As shown, our method achieves the best results on all metrics, demonstrating the effectiveness of our round-robin optimization algorithm. Figure 18 clearly shows that using a fixed order makes it difficult to optimize back to the image, especially in multi-object images. As mentioned in 3.4, optimizing features like the image background first can enhance the optimization results. Hence, Order1 performs much better than Order2 and Order3.

### D.6 INACCURATE SEGMENTATION

Figure 19 shows the reconstruction result of 3D-GOI with inaccurate segmentation. Both accurate and inaccurate segmentation can reconstruct the original image well with only minor differences, which demonstrates the robustness of our model.

### D.7 COMPUTATIONAL COSTS

We believe it is reasonable that for editing images with multiple objects in a multifaceted manner, the computational cost is positively correlated with the number of objects in the image. Furthermore, in tasks of reconstructing single objects, all our Neural Inversion encoders indeed incur more computational cost compared to the baselines E3DGE(Lan et al., 2022) and TriplaneNet(Bhattarai et al., 2023) as shown in Table 6. That is due to our goal of editing multiple objects diversely so it necessitates separate encoding predictions for various attributes of objects and backgrounds in the image, especially for affine transformation attributes, which most inversion works fail to achieve. In practice, in our experiments, the time consumed for encoding is minimal, with all codes outputted within 0.1 second. Our main time consumption is in the optimization part, but since we optimize all codes directly, even using a per-code round-robin optimization strategy is faster than the current mainstream algorithms SPI(Yin et al., 2022) and PTI(Roich et al., 2022) that require optimization of generator parameters as shown in Table 7.

## E LIMITATIONS

Despite the impressive generative capabilities of GIRAFFE, we encountered several notable issues in the tests. Notably, there was a gap between the data distribution generated by GIRAFFE and that of the original datasets, which is the main problem faced by current complex scene generation methods, making it difficult to inverse in-the-wild images. Additionally, we observed interaction effects among different codes in some of the GIRAFFE-generated images, which further complicated our inversion targets.

We believe that with the advancement of complex multi-object scene generation methods, our editing method 3D-GOI will hold immense potential for future 3D applications such as VR/AR and Metaverse.

Table 7: The comparison of hybrid-based 3D inversion methods for time costs.

| Method | time(s) |
|--------|---------|
| PTI | 55 |
| SPI | 550 |
| 3D-GOI | 30 |

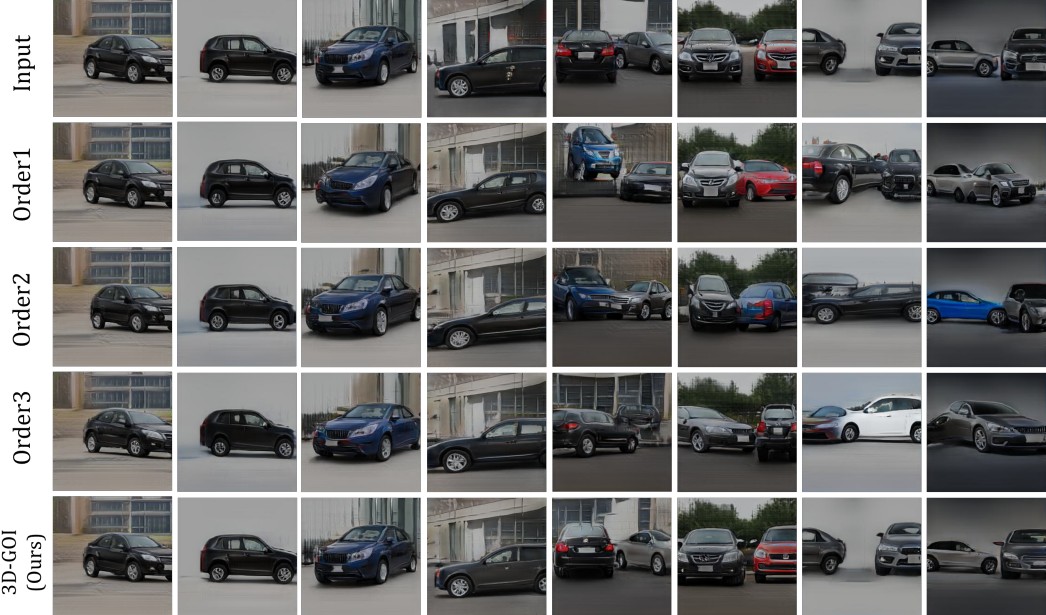

Figure 18: The figure of ablation study of the round-robin Optimization algorithm.

## F FUTUER WORK

As the first work in this new field, our current primary focus is on the accuracy of reconstruction. Our present encoding and optimization strategies are mainly aimed at achieving more precise reconstruction, while we have not given enough consideration to computational cost. Moving forward, we will continue to design the structure of the encoder to enable it to predict codes more quickly and accurately. Additionally, we need to address the entanglement issue in GIRAFFE, allowing each code to independently control the image, which may simplify our entire method process. Lastly, we need to solve the generalization issue in GAN inversion, which may require training on more real-world datasets.

## G ETHICAL CONSIDERATIONS

Generative AI models in general, including our proposal, face the risk to be used for spreading misinformation. The authors of this paper do not condone such behaviors.

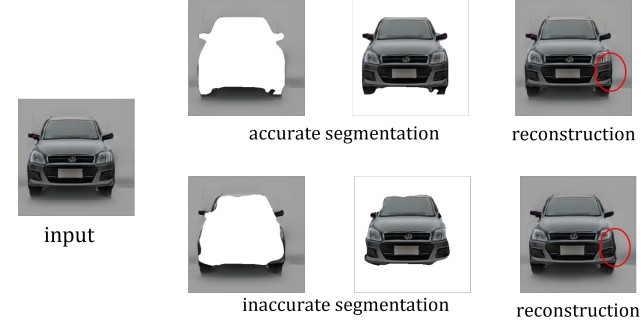

Figure 19: The figure of the reconstruction result of inaccurate segmentation.

