# OpenReview forum: "3D-GOI: 3D GAN Omni-Inversion for Multifaceted and Multi-object Editing"
_ICLR.cc/2024/Conference — Submitted to ICLR 2024_

### Official Review · Reviewer_YC1j · 2023-11-01

**Soundness:** 2 fair
**Presentation:** 2 fair
**Contribution:** 3 good
**Rating:** 5
**Confidence:** 4

**Summary:**

The current GAN inversion methods typically can only edit the appearance and shape of a single object and background while overlooking spatial information. The method proposed a 3D editing framework, 3D-GOI to enable multi-faceted editing of affine information (scale, translation, and rotation) on multiple objects.

**Strengths:**

1. 3D GAN inversion is at its preliminary start and this method is the first paper that studies multi-object 3D GAN inversion, which is an important while under-explored topic.
2. The writing is good and experiments is sound.

**Weaknesses:**

1. The overall method still lies in the hybrid optimization method, which requires code tuning after the pre-trained encoder gives a coarse estimation.
2. Lacks qualitative comparison with existing encoder-based method such as E3DGE, only Tab. 1 shows the quantitative comparisions.
3. What are the limitation of this method, and how many objects can this method handle within an image?
4. Also, any editing result based on manipulating the latent space, such as using InterfaceGAN?

**Questions:**

1. Since you allocate a single encoder to each code, what's the computational cost compared with existing encoder-based 3D GAN inversion framework such as E3DGE.
2. I really wonder why having a single optimizer to optimize all the codes in Sec.3.4 will fail, since these codes should be independent to each other?
3. How do your method address the "shape collapse" problem in 3D GAN, where a trivial 3D solution (such as a flat wall) is returned rather than a plausible 3D scene?
4. In the comparison with encoder-based 3D GAN inversion methods, TriplaneNet performs slightly better to E3DGE over MSE and LPIPS, while slightly worse on ID loss, any intuition behind it?

**Details Of Ethics Concerns:**

No.

---

> ### Author Response · Authors · 2023-11-14
> **Response to Reviewer YC1j**
>
> Thank you for your time and encouraging review! We address your questions below:
> ## Weakness:
> >1.Still lies in the hybrid optimization method.
>
> Due to the limitations of current segmentation models, achieving 100% accurate segmentation is challenging. Additionally, GIRAFFE exhibits issues with multiple attribute codes that are interdependent, relying solely on the encoder fails to obtain accurate codes. Our approach of involving more variety of codes lead to accumulated inaccurate reconstructions using only the encoder (as shown in **Appendix D.3 Figure 14**), necessitating additional optimization. Using only the optimizer significantly prolongs reconstruction time (approximately 10 minutes or more) and may fail to reconstruct images from multiple codes. Therefore, we currently consider this hybrid optimization structure as the optimal choice.
>
> >2.Lacks qualitative comparison.
>
> Please find the comparison results in **Figure 14 in Appendix D.3.**
> >3.Limitation and how many objects can this method handle.
>
> The limitations are outlined in **Appendix E**. In our experiments, we have tested with up to 6 objects. In theory, if each object in the image adheres to the distribution of the training set, our method can be used to invert and edit as many objects as needed, restricted by available computation capacity of course.
> >4.Editing result based on manipulating the latent space.
>
> We want to thank the reviewer for bringing up this more granular editing. We will definitely consider it as an improvement strategy for enhancing the generation capabilities of GIRAFFE in the future.
> ## Questions:
> >1.Computational cost compared with existing encoder-based 3D GAN inversion framework.
>
> Due to the inapplicability of existing methods to multi-object image editing, we conducted tests on the same task of synthesizing new viewpoints for faces. Compared to encoder-based methods, 3D-GOI with multiple encoders requires a total of 169M parameters and 165 GFLOPs, while E3DGE[1] has 90M parameters and 50 GFLOPs, and TriplaneNet[2] has 247M parameters and 112 GFLOPs. Inference times on a single A100 GPU are about 0.1 second for all three methods.
> |            | TriplaneNet (single encoder)| E3DGE(single encoder) | 3D-GOI(multi encoders) |
> | ---------- | --------------------- | --------------------- | ---------------------- |
> | parameters | 247M        | 90M                   | 169M                   |
> | FLOPs      | 112G        | 50G                   | 165G   |
>
> >2.Why single optimizer to optimize all the codes in Sec.3.4 will fail?
>
> As mentioned in **Appendix E**, we observed that in GIRAFFE, different object attribute codes are not entirely independent. For example, the shape code of an object can affect color control, and all attribute codes slightly impact the background as shown in **Figure 4**.   Failed cases in simultaneous optimization primarily manifest as poor background reconstruction and errors in object rotation, color, and shape features. To address this, we employ multiple optimizers to tackle these issues.
>
> >3.shape collapse
>
> This is indeed a shared challenge for most if not all existing 3D GAN generation methods. We think this is because most 3D GAN methods now use NeRF to implicitly represent the 3D space, while ensuring 3D consistency and high-fidelity image generation, may struggle with accurate modeling of visible shapes and depths. Our method, as a reverse method based on 3D GAN, therefore inevitably also faces limitations in addressing these inherent issues. We think explicit 3D representations like dense point clouds or recently proposed 3D Gaussians could potentially improve the modeling of 3D shapes while maintaining generation effectiveness.
> >4.Better MSE,LPIPS but worse  ID loss.
>
> The MSE and LPIPS metrics assess pixel-level and perceptual losses for the entire image, including the background. On the other hand, the ID loss (utilizing pre-trained Arcface for human faces and ResNet-50 for cars) evaluates object attribute similarity. As shown in **Figure 14 in Appendix D.3**, E3DGE outperforms TriplaneNet in reversing object attributes, especially in multi-car scenarios, while sometimes exhibiting weaker background inversion.
>
> Reference
> [1]Yushi Lan, Xuyi Meng, Shuai Yang, Chen Change Loy, and Bo Dai. Self-supervised geometry-aware encoder for style-based 3d gan inversion. In Proceedings of the IEEE/CVF Conference on Computer Vision and Pattern Recognition, pp. 20940–20949, 2023.
> [2]Ananta R Bhattarai, Matthias Nießner, and Artem Sevastopolsky. Triplanenet: An encoder for eg3d inversion. arXiv preprint arXiv:2303.13497, 2023.

---

### Official Review · Reviewer_gkt1 · 2023-11-01

**Soundness:** 3 good
**Presentation:** 2 fair
**Contribution:** 2 fair
**Rating:** 5
**Confidence:** 4

**Summary:**

The study presents 3D-GOI, an innovative 3D editing framework that seeks to provide multifaceted editing capabilities for multiple objects in a given scene. 3D-GOI employs a three-stage approach that first segments the objects and background in a multi-object image, then uses a Neural Inversion Encoder to derive coarse attribute codes for each object, and finally, through a round-robin optimization strategy, refines these codes to recreate the image. Leveraging GIRAFFE, a well-known 3D GAN, the method allows for detailed editing on both object and scene scales, including adjustments to object appearance, position, and even the camera pose.

**Strengths:**

1. The paper's emphasis on multi-object and multifaceted editing not only differentiates it but also highlights the immense potential such editing holds for future technologies.
2. The ablation study breaks down different components of the proposed method, such as the Neural Inversion Encoder and the Round-robin Optimization algorithm, shedding light on their individual contributions.

**Weaknesses:**

The primary concern with the paper lies in its dependency on multiple stages and components for accuracy, which hints at potential inefficiencies and complexities in the method. Specifically, the Neural Inversion Encoder's inability to precisely predict codes independently underscores a fundamental limitation. This necessitates the round-robin optimization approach, adding another layer of complexity. Furthermore, the model's significant deviations in predicting background codes for multi-object scenes highlight challenges in accurately reconstructing detailed and complex backgrounds. Additionally, the model's sensitivity to the order of feature optimization raises concerns about its consistency and robustness in various scenarios. Coupled with its reliance on pre-trained models, these issues collectively indicate potential areas that could hinder the model's broad applicability and efficiency.

1. With multiple dedicated encoders for fine-tuning specific attributes, there's a risk of the model fitting too closely to training data, resulting in poor generalization to unseen data. It would be valuable to see empirical evidence showcasing the model's performance on diverse, unseen data.
2. Differentiating between closely related attributes, such as scaling and translation, can be tricky. If the encoders aren't adequately designed to distinguish between such nuances, they might produce overlapping or redundant codes. This can hinder precise scene reconstruction and may result in ambiguity when trying to decipher the attributes of a given scene. It would be beneficial for the authors to offer visualizations of the latent space, as this could shed light on potential overlaps and provide deeper insights into the latent representations.

3. The model's performance is highly contingent on accurate segmentation. If the segmentation phase is off even slightly, the downstream processes, including scene representation and attribute extraction, can be significantly affected. This makes the model sensitive to the initial stages of processing, potentially reducing its resilience against imperfect inputs. It would be beneficial to see results or demonstrations of how the model performs with imperfect segmentation inputs, providing a more holistic view of its resilience.

**Questions:**

1. How can the Neural Inversion Encoder's accuracy be improved, so it becomes less dependent on subsequent optimization for precise code predictions?

2. Given the two-step approach of coarse estimation and precise optimization, what are the computational costs associated with the 3D-GOI method, and how do they compare with other inversion methods?

3. Beyond the potential fields mentioned in the conclusion, are there other applications where 3D-GOI could be particularly useful?

---

> ### Author Response · Authors · 2023-11-15
> **Response to Reviewer gkt1**
>
> Thank you for your time and encouraging review! We address your questions below:
> ## Weakness:
> Thank you for this valuable feedback. Indeed, encoders that can capture precise codes  wouldn't require additional optimization steps. Unfortunately, all mainstream GAN inversion encoders fail to accurately predict all codes in settings with multiple objects and numerous codes, as shown in **Figure 14** . As we discussed in the limitations, we found that in GIRAFFE, the attribute codes of different objects are not entirely independent. For instance, the shape code of an object can influence its color, and all attribute codes slightly affect the background as shown in **Figure 4**. Therefore, as the number of objects (and hence codes) increases, their interrelationships become more complex. Meanwhile the accuracy of encoder predictions varies across different attributes; yet only with precise predictions for all attributes can we accurately reconstruct the image.  Therefore, using just the segmented images of objects seem to become a dead end for obtaining precise codes. As such, we propose to use the round-robin optimization algorithm to accurately determine the code values.
> >1.poor generalization to unseen data.
>
> In our tests on  unseen vehicle images, we found that 3D-GOI performs well in inversing and multifaceted editing vehicles but faces challenges in reconstructing the background, resulting in less ideal background reconstruction.  This is due to the difference between the background distributions of the dataset and the real-world. We plan to incorporate methods like optimizing the generator parameters on parallel, similar to PTI, in the future work.
>
> >2.Differentiating and visualizaing between closely related attributes
>
> Thank you very much for your suggestion. Visualizing all the codes is challenging for us because codes for different attributes reside in separate latent spaces. Additionally, GIRAFFE couples them together using affine transformations and a NeRF neural network, making it much more complex than the typical GAN process of mapping from latent space to images. We will consider this as part of our future work.
>
> >3.inaccurate segmentation
>
> We tested an inaccurate segmentation model with an AP50 of 0.7 (where the current SOTA is 0.8 in InternImage-H) as shown in **Appendix D.6** and **Figure 19**. Despite the coarse codes being less accurate, our round robin optimization algorithm was still able to refine them effectively.
>
> ## Questions:
> >1.How can the Neural Inversion Encoder's accuracy be improved?
>
> Improving the accuracy of the Neural Inversion Encoders  is challenging, as our current architecture has undergone multiple refinements, confirmed by experiments (as shown in **Table 1** and **Figures 14, 15**) to show our encoder outperform the baselines. Our current multi-encoder structure, although using multiple encoders, achieves the highest accuracy.
> >2.computational costs
>
> Under the same experimental setup, compared to encoder-based inversion methods, 3D-GOI with multiple encoders requires a total of 169M parameters and 165G FLOPs, while E3DGE[1] has 90M parameters and 50G FLOPs, and TriplaneNet[2] has 247M parameters and 112G FLOPs. All three methods achieve inference times of approximately 0.1 second on a single A100 GPU.Compared to optimization-based methods, 3D-GOI's optimization process takes 30 seconds, while SPI[3] requires 550 seconds, and PTI[4] requires 55 seconds.
> || SPI  | PTI  | 3D-GOI |
> |----------------- | ----- | ------ | ------ |
> | optimization time | 550s | 55s  | 30s    |
>
> || TriplaneNet (single encoder)| E3DGE(single encoder) | 3D-GOI(multi encoders) |
> |---------- | ----------- | --------------------- | ---------------------- |
> | parameters| 247M | 90M| 169M|
> | FLOPs   | 112G | 50G| 165G  |
> >3.other applications where 3D-GOI could be particularly useful.
>
> Yes, 3D muli-object editing has broad applications. It can enhance film production, video game development, product design, interior design, and architecture by enabling realistic visual content creation and edition. Artists and content creators can leverage this framework to craft personalized and innovative 3D art and media.
> Reference
> [1]Yushi Lan, Xuyi Meng, Shuai Yang, Chen Change Loy, and Bo Dai. Self-supervised geometry-aware encoder for style-based 3d gan inversion. In Proceedings of the IEEE/CVF Conference on Computer Vision and Pattern Recognition, pp. 20940–20949, 2023.
> [2]Ananta R Bhattarai, Matthias Nießner, and Artem Sevastopolsky. Triplanenet: An encoder for eg3d inversion. arXiv preprint arXiv:2303.13497, 2023.
> [3]Fei Yin, Yong Zhang, Xuan Wang, Tengfei Wang, Xiaoyu Li, Yuan Gong, Yanbo Fan, Xiaodong Cun, Ying Shan, Cengiz Oztireli, et al. 3d gan inversion with facial symmetry prior. arXiv preprint arXiv:2211.16927, 2022.
> [4]Daniel Roich, Ron Mokady, Amit H Bermano, and Daniel Cohen-Or. Pivotal tuning for latent-based editing of real images. ACM Transactions on Graphics (TOG), 42(1):1–13, 2022.

---

### Official Review · Reviewer_f1rX · 2023-11-02

**Soundness:** 3 good
**Presentation:** 2 fair
**Contribution:** 3 good
**Rating:** 6
**Confidence:** 3

**Summary:**

This paper presents a novel inversion framework to invert the latent codes from GAN for 3D editing on the basis of GIRAFFE. It focuses on inverting the multiple codes of multiple objects in a single image, where the major challenges are two-fold: 1. latent code disentanglement; 2. effective optimization method to solve them. To address the two targets, the authors respectively propose the Neural Inversion Encoder, and Round-robin Optimization, which I believe are the major contributions. The method follows the scene decomposition method similar to GIRAFFE. After decomposition, the Neural Inversion Encoder is proposed to do coarse estimation for each object property (e.g., appearance, shape, etc.). To better optimize so many latent codes from multiple objects, they designed the Round-robin Optimization to update the gradients with a loss gradient ranking style.

**Strengths:**

This paper focuses the task of GAN inversion problem of different latent codes from multiple objects in a single image. To address the challenges lying in the multiple latent code estimation and optimization, there are two technical strengths:

1. The round-robin Optimization strategy to optimize all codes simultaneously.
2. the Neural Inversion Encoder to encode each code for initial estimation.
3. Extensive experiments that demonstrate their effectiveness

**Weaknesses:**

The weaknesses in this paper are also obvious.

1. This paper still follows the framework of GIRAFFE by using its scene decomposition manner and training strategy, even though the authors claim that they have more object properties to encode.

2. Second paragraph of Intro: there are some other methods based on VAEs (Sync2Gen, ICCV'21) and transformers (NeurIPS'21)

3. Sec 3.2,  and Intro, there is no need to have so many texts to elaborate on why you use the segmentation method. It is pretty intuitive.

4. During training, you mentioned that you train an encoder for one code at a time, and keep the other codes at their true values (Sec 3.3). In Round-robin Optimization (Sec 3.4), you also mentioned that you prioritize optimizing the code with decreasing loss. IIf it does not decrease, then change to optimize the other codes. How do you combine the two training strategies?

5. The paper should be written in a more concise way (see 3.). There are many important sections that should not be moved to the supplementary, e.g., related works, and qualitative comparisons with baselines.

**Questions:**

1. Since the authors claim that they use an optimizer for each code, and the Round-robin Optimization method optimizes each code in turn, what is the time efficiency of this method compared with the baselines?

---

> ### Author Response · Authors · 2023-11-14
> **Response to Reviewer f1rX**
>
> Thank you for your time and encouraging review! We address your questions below:
> ### Weakness：
> >1.This paper still follows the framework of GIRAFFE.
>
> Our work fundamentally differs from GIRAFFE which generates images by initializing random codes as input, while our method takes multi-object scene images as input and map them back to the GIRAFFE code space. Therefore, although some parts of our method resemble GIRAFFE, there are fundamental mechanism differences.
> >2.Second paragraph of Intro: there are some other methods based on VAEs (Sync2Gen, ICCV'21) and transformers (NeurIPS'21)
>
> Thank you for this suggestion. We have added the discussion of mentioned references in the introduction section.
> >3.The segmentation method part  is pretty intuitive.
>
> Thank you for this suggestion. We will shorten the relevant text in the final version.
> >4.How do you combine the two training strategies?
>
> Our approach first trains the Neural Inversion Encoders (NIE) for each object. Afterwards, we input the coarse code generated by the frozen NIEs to the optimizer and use the round-robin optimization algorithm to refine them into accurate codes.
>
> >5.The paper should be written in a more concise way.
>
> Thank you for this suggestion. We will streamline Section 3, including related work, more details of the encoder design, and the experimental part in the main text.
> ### Questions:
> > 1.Time efficiency of this method compared with the baselines.
>
> Our method is very time-efficient compared to  methods that need optimization. For instance, to invert a single image, SPI[1] requires 550 seconds of optimization, PTI[2] requires 55 seconds and our 3D-GOI only needs 30s. This is primarily because, unlike other approaches, we only optimize the codes but not the generator parameters.
>
> |                   | SPI  | PTI  | 3D-GOI |
> | ----------------- | ------| ------ | -------- |
> | optimization time | 550s | 55s  |   30s    |
>
> Reference
> [1]Fei Yin, Yong Zhang, Xuan Wang, Tengfei Wang, Xiaoyu Li, Yuan Gong, Yanbo Fan, Xiaodong Cun, Ying Shan, Cengiz Oztireli, et al. 3d gan inversion with facial symmetry prior. arXiv preprint arXiv:2211.16927, 2022.
>
> [2]Daniel Roich, Ron Mokady, Amit H Bermano, and Daniel Cohen-Or. Pivotal tuning for latent-based
> editing of real images. ACM Transactions on Graphics (TOG), 42(1):1–13, 2022.

---

### Author Response · Authors · 2023-11-16
**General Response**

Dear reviewers and AC,
We sincerely appreciate your valuable time and effort spent reviewing our manuscript.

As reviewers highlighted, our work is the first paper that studies multi-object 3D GAN inversion (Reviewer YC1j) with effective method(Reviewer f1rX),strong empirical results (ALL Reviewers), and great application prospects (Reviewer gkt1).

We appreciate your constructive comments on our manuscript. In response to the comments, we have carefully revised and enhanced the manuscript with the following additional discussions and experiments:
+ Additional experiment of inaccurate segmentation as input(Appendix D.6,Figure 19)
+ Additional experiment of computational and time costs (Appendix D.7,Table 6,7)
+ Added two references based on VAE and Transformer generation methods.(Section 1)
+ Discussion about the future work(Appendix F)

These updates are temporarily highlighted in“$\textcolor{blue}{blue}$” for your convenience to check.

We hope our response and revision sincerely address all the reviewers’ concerns.

Thank you very much.

Best regards,
Authors.

---

### Author Response · Authors · 2023-11-21
**Looking Forward to Your Reply**

Dear reviewers,

We sincerely value and appreciate your insightful review. Your queries have been meticulously evaluated, and we've endeavored to offer comprehensive responses. We cordially invite you to check our replies, hoping they resonate with your perspectives. Your dedication and expertise in reviewing our work greatly help our work to improve.

Warm regards,
The Authors

---

### Meta-Review · Area_Chair_2p9x · 2023-12-06

**Metareview:**

This paper presents a 3D editing framework that enables multifaceted editing of affine information on multiple objects. The framework focuses on inverting multiple codes of multiple objects in a single image where Neural Inversion Encoder is developed for coarse code of each object and Round-robin Optimization is used to obtain precise codes to reconstruct the image.  Although the current GAN inversion methods can only edit the appearance and shape of a single object and background, the proposed method enables multifaceted editing on multiple objects.  This research direction has potential such editing holds for future technologies.  On the other hand, the reviewers raised concerns regarding lack of in-depth discussion about the proposed method such as dependency on multiple stages and components for accuracy, two-stage training strategy, and generalization to diverse, unseen data.  Impact of inaccurate segmentation was also raised as a concern.  More concise writing was pointed out as well. The authors’ rebuttal resolved the raised concerns to some extent; however, the main concerns still remain.  For example, dependency of the proposed method on multiple stages and components for accuracy was not addressed. The sensitivity to the order of feature optimization was not addressed, either.  This concern leads to consistency and robustness of outputs by the proposed method in various scenarios.  Two-stage training strategy (hybrid optimization) should be discussed from different perspectives in more detail.  Properly addressing these concerns will give better understanding significance of the proposed method.  Addressing them is the authors’ responsibility, so that the claims of the paper are properly justified.  Finally, the revised manuscript was not written concisely; just simple results on inaccurate segmentation and computational cost were added in the appendix. The proposed interesting method should be appreciated, but without substantial revision which requires at least one more round of peer review, the paper cannot be accepted.

**Justification For Why Not Higher Score:**

N/A

**Justification For Why Not Lower Score:**

N/A

---

### Decision · Program_Chairs · 2024-01-16

Reject